# Belief updating in bipolar disorder predicts time of recurrence

**Paolo Ossola[1]\*, Neil Garrett[2], Tali Sharot[3], Carlo Marchesi[1]**

[1]Psychiatry Unit, Department of Medicine and Surgery, Università di Parma, Parma, Italy; [2]Department of Experimental Psychology, University of Oxford, Oxford, United Kingdom; [3]Affective Brain Lab, Department of Experimental Psychology, University College London, London, United Kingdom

**Abstract** Bipolar disorder is a chronic relapsing condition in which mood episodes are interspersed with periods of wellbeing (euthymia). Shorter periods of euthymia are associated with poorer functioning, so it is crucial to identify predictors of relapse to facilitate treatment. Here, we test the hypothesis that specific valence-dependent learning patterns emerge prior to the clinical manifestation of a relapse, predicting its timing. The ability to update beliefs in response to positive and negative information was quantified in bipolar patients during euthymia, who were then monitored for 5 years. We found that reduced tendency to update beliefs in response to positive relative to negative information predicted earlier relapse. Less updating in response to positive information may generate pessimistic beliefs, which in turn can lead to more severe prodromal symptoms (e.g. sleep disturbance, irritability etc.). The results suggest that measuring valence-dependent belief updating could facilitate risk prediction in bipolar disorder.

## Introduction

Bipolar disorder is estimated to affect approximately 3% of the general population (*American Psychiatric Association, 2013*). The disorder is characterized by successive periods of elation (mania) and depression interspersed with periods of euthymia, an asymptomatic phase in which patients are in clinical remission (*Grande et al., 2016*). Shorter periods of euthymia between relapses are associated with poorer functioning, increased odds of suicidality, disability, unemployment, and hospitalization (*Peters et al., 2016*). A major research goal is thus to identify predictors of upcoming relapse in order to facilitate timely treatment (*Harrison et al., 2016*). Yet, predicting relapse using existing clinical diagnostic tools or demographic information has proven largely ineffective in bipolar disorder (*Bukh et al., 2016*; *de Dios et al., 2012*; *Vieta et al., 2018*).

Here, we test the hypothesis that prior to the clinical manifestation of a relapse patients exhibit a specific learning pattern from self-relevant information, which makes them more vulnerable to relapse. In particular, it has been shown that healthy individuals tend to update their beliefs to a greater extent in response to unexpected positive information (e.g. learning that the likelihood of receiving a job offer is greater than expected) than negative information (e.g. learning it is lower than expected; *Kappes et al., 2020*; *Kuzmanovic et al., 2016*; *Ma et al., 2016*; *Moutsiana et al., 2015*; *Sharot et al., 2011*). Such asymmetry in valence-dependent belief updating leads to optimistic beliefs (*Sharot et al., 2011*). This is because a person who takes in positive information to a greater extent than negative information from their experiences in the world, uses such information to update their beliefs, which eventually leads to more positive expectations.

In contrast, individuals with depression do not show a positivity bias in learning (*Garrett et al., 2014*; *Korn et al., 2014*). These patients update their beliefs less in response to positive information (*Korn et al., 2014*) and more in response to negative information (*Garrett et al., 2014*), leading to pessimistic views that may exacerbate symptoms.

**\*For correspondence:**
paolo.ossola@unipr.it

**Competing interests:** The authors declare that no competing interests exist.

However, it is unknown whether a lack of positivity bias in learning *precedes* symptoms of affective disorders. If it does, then measuring learning biases could facilitate early diagnosis. Indeed, there is theoretical basis to believe that reduced learning from positive information and/or enhanced learning from negative information will occur before a relapse. In particular, a reduction in learning from positive information (or enhancement in response to negative information) will cause individuals to take in less desirable information from the world around them (or more undesirable information), which over time will tip beliefs toward a pessimistic direction. Once a patient generates a negative set of beliefs, deleterious symptoms may appear such as negative mood, sleep disturbance, suicidal thoughts etc.

Testing this possibility on Bipolar patients is especially compelling for three reasons. First, bipolar patients tend to relapse frequently, with a mean distance between episodes of 18 months on average (*Radua et al., 2017*). If we were to examine how patients update their beliefs during euthymia to predict how soon they would relapse, within only a few years most patients would have likely relapsed, which would allow us to test the reliability of those predictions. Second, whilst traditional diagnostic tools that measure symptom severity has proven useful in predicting relapse in unipolar depression (*Dinga et al., 2018*), these are largely ineffective in bipolar disorder (*Bukh et al., 2016*; *de Dios et al., 2012*). Third, by examining bipolar patients we could test whether valence-dependent belief updating predicts episodes of both polarities (that is mania and depression) differently or in the same manner. On one hand, the one-dimensional model suggests that mania and depression lie on opposite ends of the same mood spectrum. Hence manic or depressive symptoms arise only in the absence of the other (*Bonsall et al., 2015*; *Bonsall et al., 2012*; *Chang and Chou, 2018*; *Daugherty et al., 2009*; *Eldar et al., 2018*; *Eldar et al., 2016*; *Eldar and Niv, 2015*; *Mason et al., 2017*; *Nana, 2009*; *Steinacher and Wright, 2013*). We may thus expect that positivity bias in belief updating would predict manic episodes and vice versa for depressive episodes. Other models, however, conceptualize mania and depression as independent dimensions (*Bystritsky et al., 2012*; *Cochran et al., 2018*; *Goldbeter, 2013*; *Goldbeter, 2011*; *Hadaeghi et al., 2015*; *Huber et al., 2000*; *Lopez, 2008*). According to these models, perturbations can trigger episodes of either polarity. The degree to which patients update beliefs in response to positive and negative information may reflect such perturbations (or perhaps even cause them) predicting relapse of both polarities similarly.

To test whether a valence-dependent bias in belief updating, or the lack thereof, precedes relapse of bipolar episodes, we tested patients with bipolar disorder on the belief update task (*Sharot et al., 2011*) while they were in the euthymic stage. The patients were then followed up for 5 years and their time of relapse was recorded.

## Results

### Belief update task

Forty-five patients diagnosed with bipolar disorder (see *Supplementary file 1* for patients' demographics and characteristics) performed the belief update task while in the euthymic phase (*Figure 1*). Nine patients dropped out of the study before 5 years elapsed. The rest - 36 patients - were monitored for symptoms approximately every 2 months for the following 5 years. The task allows quantification of belief change in response to information that is better or worse than expected. The task and analysis employed here have been used numerous times before (*Chowdhury et al., 2014*; *Garrett et al., 2018*; *Garrett et al., 2014*; *Garrett and Sharot, 2016*; *Garrett and Sharot, 2014*; *Kappes et al., 2018*; *Korn et al., 2014*; *Kuzmanovic et al., 2016*; *Kuzmanovic et al., 2014*; *Moutsiana et al., 2013*; *Moutsiana et al., 2015*; *Sharot et al., 2012b*; *Sharot et al., 2012a*; *Sharot et al., 2011*). Participants were presented with 40 adverse life events (e.g. robbery, card fraud) and asked to estimate how likely the event was to happen to them in the future (this is referred to as the *first estimate*). They were then presented with the base rate of the event in a demographically similar population (this is referred to as *information*). For each trial, an *estimation error* term was calculated as the difference between the probability presented (information) and participants' first estimate on that trial. In a second session, participants were asked again to provide estimates of their likelihood of encountering the same events (this is referred to as *second estimate*). Trials were divided into those in which participants received *good news* (i.e. the probability

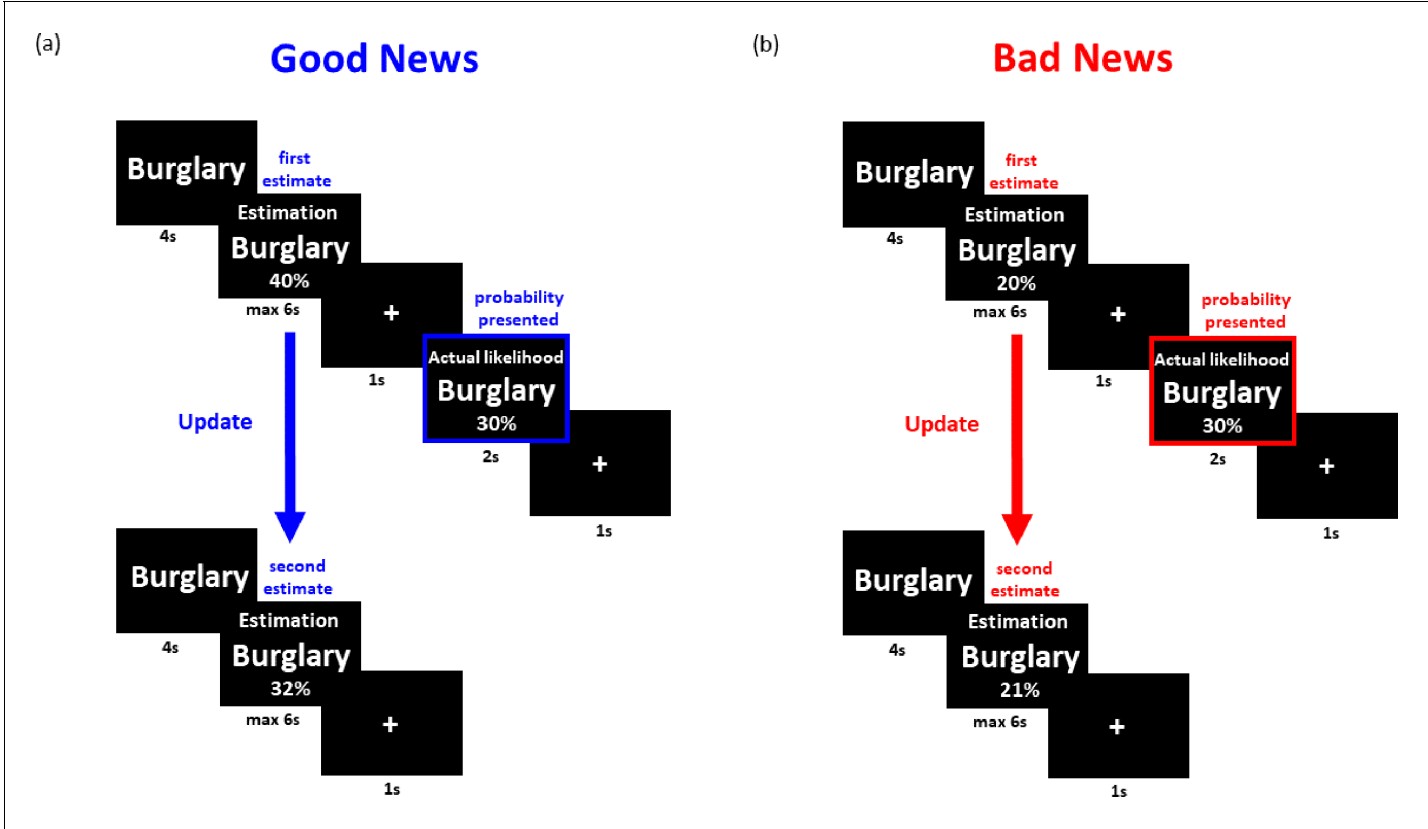

**Figure 1.** Belief update task. In the first session (top row), on each trial, participants were presented with a short description of an adverse life event and asked to estimate how likely this event was to occur to them in the future. They were then presented with the probability of that event occurring to someone from the same age, location, and socio-economic background as them. After completing 40 such trials they then completed the second session (bottom row). This was the same as the first except that the average probability of the event to occur was not presented. Shown are examples of trials for which the participant's estimate was higher or lower than the statistical information provided leading to receipt of (a) good news and (b) bad news, respectively. Update is calculated as the difference between participants estimates in the two sessions (i.e. pre- and post-information). The blue and red boxes are here to illustrated good and bad news, and did not appear in the actual experiment.

presented of encountering an aversive event was lower than the subject's first estimate of their own probability; see *Figure 1a*) or *bad news* (i.e. the probability presented was higher than the subject's first estimate of their own probability; see *Figure 1b*). While information can be better or worse than expected, all stimuli are negative life events. Thus, comparison is never between positive and negative stimuli, but between information that is better (thus subjectively positive) or worse (thus subjectively negative) than expected. Different methods of dividing trials in this task has shown to yield similar results (*Garrett and Sharot, 2014*), thus we used the original task design here.

Participants change in beliefs for each trial is referred to as the **update** and is calculated for good news trials as the difference in participants' first estimate and second estimates (i.e. pre- and post-information), whereas for bad news trials as the difference between the participants' second estimate and first estimate. Thus, in both cases, positive numbers indicate a move toward the information.

For each subject, we then separately calculated the average update for good news and bad news trials. The difference between the two types of trials is the '*Update Bias*', an explicit measure of valence-dependent belief updating. Positive update bias scores indicate greater updating in response to good news relative to bad news (this is an optimistic update bias) whilst negative scores indicate the reverse (a pessimistic update bias).

At the end of the task, participants were asked to provide the actual probability previously presented for each event. *Memory errors* were calculated as the absolute difference between the probability previously presented and the participants' recollection of that statistic. Participants also provided a number of ratings for each event on a likert scale according to: past experience,

familiarity, negativity, vividness and emotional arousal (we call these '*subjective ratings*', see Materials and methods).

The question we address here is whether the magnitude of the bias in belief updating (Update Bias) is associated with the patients' prognosis. Patients were followed up for 5 years after they completed the update bias task and we logged their episode recurrence during this period. This enabled us to calculate the duration of time spent in euthymia for each participant from when they completed the task in a euthymic phase until their next episode ('*future time in euthymia*'). We then examined if the future time in euthymia was predicted by their update bias score.

## Belief update bias is associated with future duration of euthymia

We found an association between the update bias and future time in euthymia with a larger positive update bias related to longer time in euthymia. In particular, we implemented the linear regression with future time in euthymia as the dependent measure and update bias (our main measure of interest) as the independent measure. We also added (1) prior beliefs (mean first estimate) and the difference between good news trials and bad news trials on the following measures to control for possible confounds: (2) estimation errors (3) memory error (4) number of trials (5-6) reaction times at the first and second estimate and (7-11) all subjective ratings (that is ratings on familiarity, prior experience, vividness, emotional arousal and negativity) (see Materials and methods). The analysis revealed a significant effect of belief update bias (Beta = 0.589, p=0.004, bootstrapping on 10,000 samples, 95% CI = 0.122–1.129) with a larger bias predictive of greater future time in euthymia (*Figure 2*, *Supplementary file 2a*). To test whether this effect may have been attributed to multicollinearity within the model, we calculated the Variance Inflation Factor (VIF) of the update bias. This revealed a low VIF (=1.93), which indicates that the variable does not display collinearity with the other variables in the model. Lower first estimation (optimistic priors) was also associated with longer future time in euthymia (Beta = −2.74, p=0.003) along with the difference in the number of good and bad news trials (Beta = 2.02, p=0.003).

To further test the robustness of the effect we used the 'model averaging' approach (*Freckleton, 2011*). This approach involves first running every single combination of models given the independent variables. For example, running a model only with two of the variables, only three, only four and so on. Each time with a different combination of independent variables. Our original model included 13 variables, thus this involved running 4095 nested models. Then the betas of each variable are averaged across all models, weighting them on the model's BIC (Bayesian Information Criterion) (*Freckleton, 2011*). This exercise revealed a significant effect of update bias in explaining time in euthymia (weighted estimates = 0.47, 95% CI = 0.34, 0.60). Moreover, the eight best fitting models (out of 4095), according to the lowest BIC score, all included the update bias.

To test whether the update bias predicted future time in euthymia differently for manic and depressive episodes, we added to the linear regression above the polarity of the next episode and its interaction with the update bias as predictors. Neither the next polarity (Beta = 0.242; p=0.233) nor the interaction were significant predictors (Beta = −0.72, p=0.803), while the update bias remained a significant predictor (Beta = 0.490; p=0.052). This suggests that the update bias does not predict future time in euthymia differently based on the polarity of the future episode.

Finally, we examined if the update bias predicts time in euthymia while only controlling for variables for which significant differences exists between good news trials and bad news trials, as those differences may introduce noise that can obscure the effects of update bias if not controlled for. To identify these variables, we tested with paired sample t-test if any of the variables in Model 1 showed significant differences between good and bad news trials. This revealed significant effects for each subjective rating: past experience, familiarity, vividness, arousal, vividness and negativity (vividness t(34) = 3.96, p<0.001; familiarity t(34) = 3.38, p=0.002; prior experience t(34) = 3.304, p=0.002; emotional arousal t(34) = 3.971, p<0.001; negativity t(34) = 2.414, p=0.021). No other differences were found (*Supplementary file 1a*). We thus ran a linear regression with future time in euthymia as the dependent measure and update bias (our main measure of interest) as the independent measure controlling only for the subjective measures above. We found that Update Bias was significantly associated with time in euthymia (Beta = 0.351; p=0.047), with no multicollinearity (VIF = 1.098). In this model, ratings of emotional arousal (Beta = 0.775, p=0.016) and negativity (Beta = −0.576, p=0.026) were also significant predictors. Note, however, that these latter effects

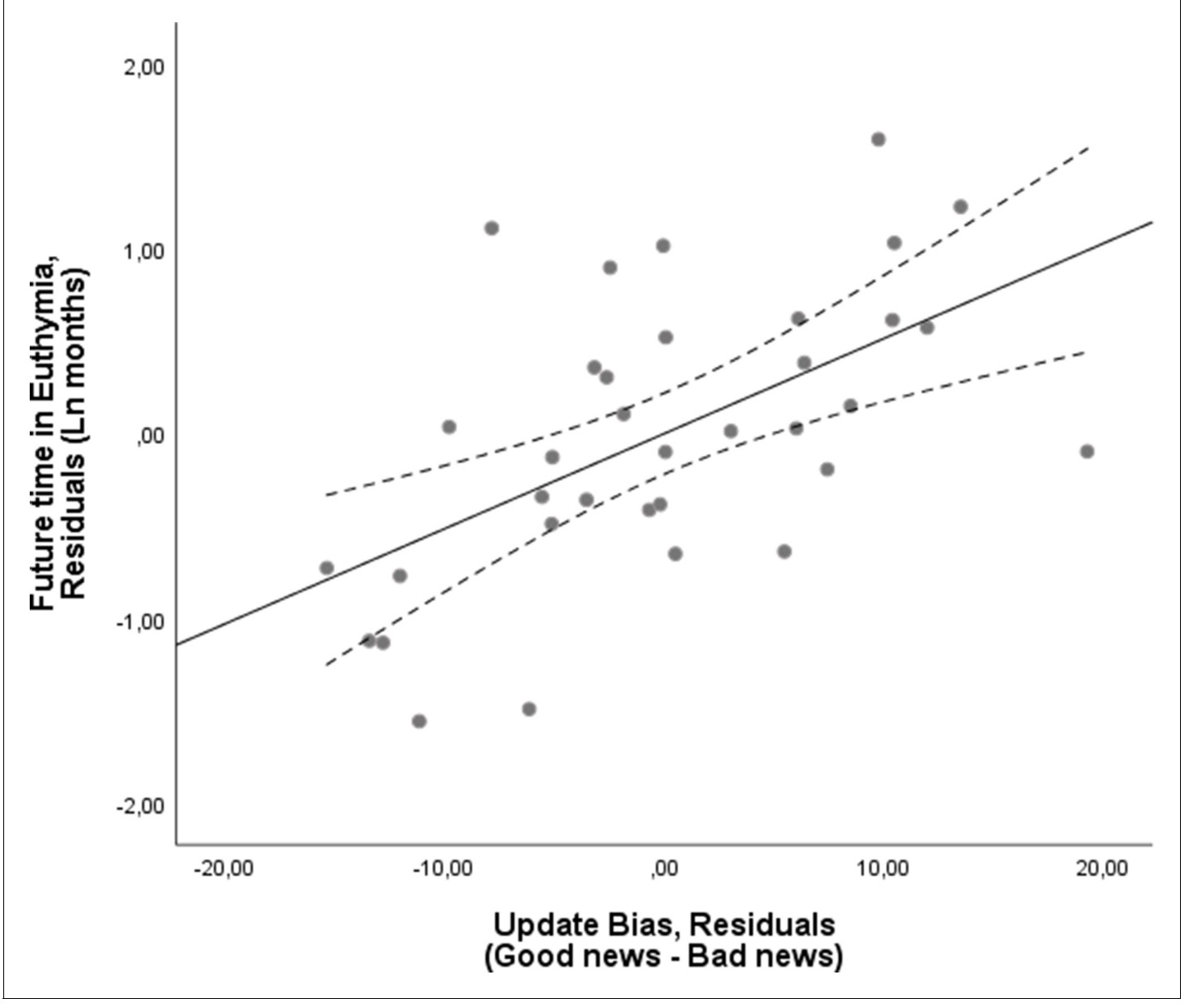

**Figure 2.** Larger update bias is associated with longer future time in euthymia. Larger optimistic update bias at test is associated with future time in euthymia. The X-axis represents the residuals in predicting the update bias from all the independent variables included in the model as a control. These are the subjects' prior beliefs represented by their mean first estimate and the difference between good news trials and bad news trials on estimation errors, memory error, number of trials, reaction times at the first and second estimates and all the subjective ratings. The Y-axis represents the residuals from predicting time in euthymia (logarithmic transformed to account for skewedness) from the same independent variables, that is the full model without the Update Bias. The dotted lines represent 95% confidence intervals.

do not remain significant in the original model once controlling for other task variables whilst the update bias does.

## Future time in euthymia is associated with greater updating in response to good news and less updating in response to bad news

Thus far, we have shown that patients with a greater, more optimistic, update bias experienced euthymia for longer, whilst those with a smaller bias relapsed faster. Since the update bias is a difference score (the difference between updating beliefs in response to good and bad news), the association could be due to a variation in belief updating in response to good news, bad news or both. To

that end, we ran two separate linear regressions predicting future time in euthymia. These were exactly as described above except that in one (Model 1a), update in response to good news was an independent variable in place of update bias. In the other (Model 1b), update in response to bad news was the independent variable in place of update bias. We found that longer future time in euthymia was associated with greater updating from good news (Model 1a: good news

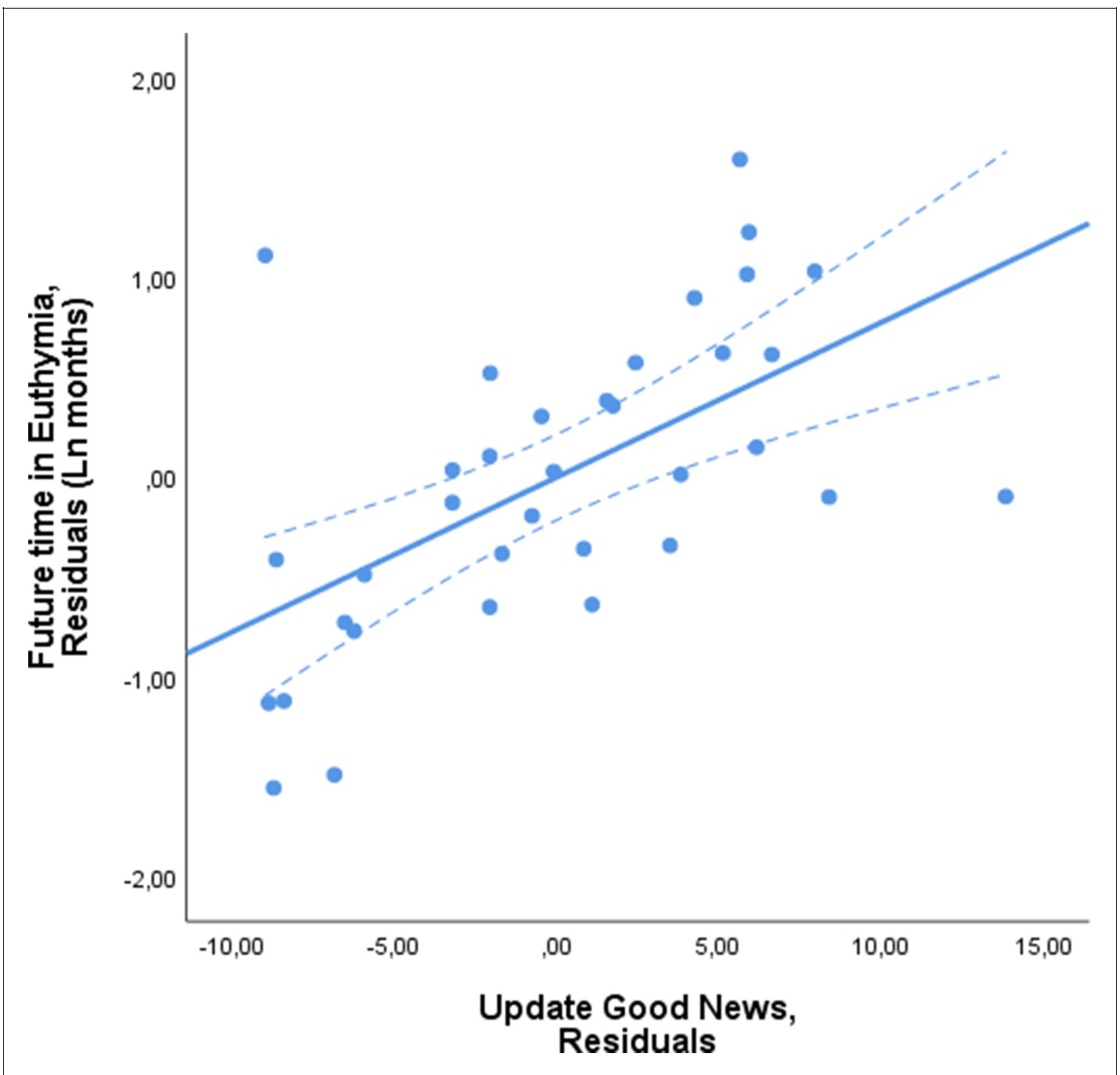

**Figure 3.** Future time in euthymia is related to greater belief updating in response to good news. Future time in euthymia was positively related to updating in response to good news. The X-axis represents the residuals in predicting the Update in response to good news from all the independent variables included in the model as a control. These are the subjects' prior beliefs represented by their mean first estimate and the difference between good news trials and bad news trials on estimation errors, memory error, number of trials, reaction times at the first and second estimate and all the subjective ratings. The Y-axis represents the residuals from predicting time in euthymia (logarithmic transformed to account for skewedness) from the same independent variables. The dotted lines represent 95% confidence intervals.

Beta = 0.570, p=0.002, VIF = 1.673, *Figure 3*, *Supplementary file 2b*). Updating from bad news (Model 1b) showed a non-significant association in the opposite direction (Beta = −0.276, p=0.219, VIF = 2.038, *Figure 4*, *Supplementary file 2c*). When including both update for good news and bad news in the same linear regression, update for good news was again significantly associated with future time in euthymia (Beta = 0.543, p=0.004, VIF = 1.721) whilst update from bad news was not (Beta = −0.175; p=0.354, VIF = 2.097).

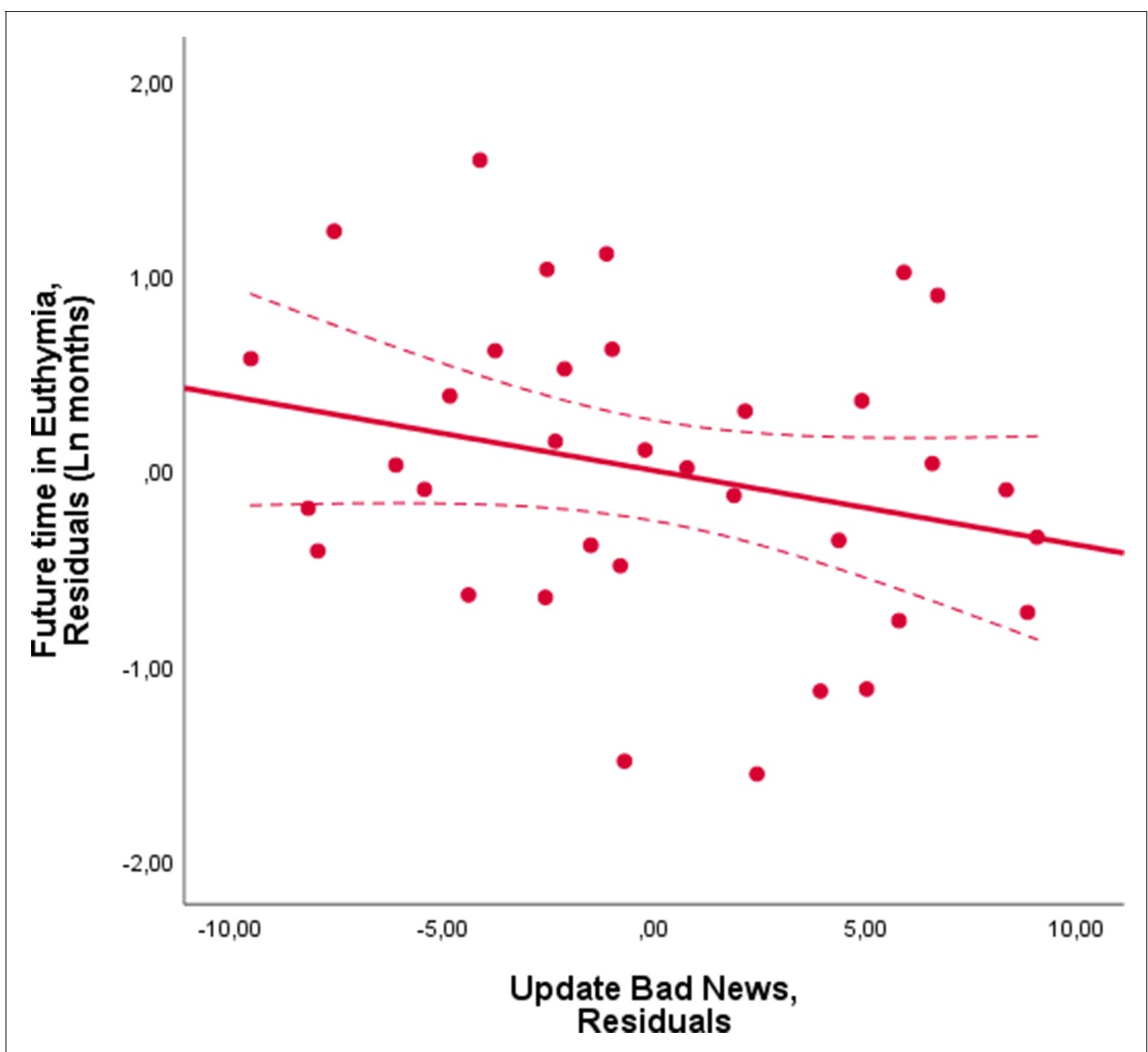

**Figure 4.** Future time in euthymia is related to smaller belief updating in response to bad news. Future time in euthymia showed a negative, non-significant, association with updating in response to bad news. The X-axis represents the residuals in predicting the Update score in response to bad news from all the independent variables included in the model as a control. These are the subjects' prior beliefs represented by their mean first estimate and the difference between good news trials and bad news trials on estimation errors, memory error, number of trials, reaction times at the first and second estimate and all the subjective ratings. The Y-axis represents the residuals from predicting time in euthymia (logarithmic transformed to account for skewedness) from the same independent variables. The dotted lines represent 95% confidence intervals.

## Update bias is associated with future time in euthymia beyond all other clinical indicators

To examine if traditional clinical indicators also predicted future time in euthymia, we ran the same linear regression again (including all task controls as described above), but this time we included clinical and demographic variables. Specifically: age, education, gender, bipolar type (I or II), being on mood stabilizers (lithium, anticonvulsants), antipsychotics, antidepressants, history of psychotic symptoms, duration of illness and Beck Depression Inventory scores (*Altamura et al., 2015*; *Lähteenvuo et al., 2018*; *Radua et al., 2017*; *Tundo et al., 2018*).

Update Bias was again strongly related to future time spent in euthymia (Beta = 0.741, p=0.001). The VIF score again indicated no multicollinearity between update bias and the other variables (VIF = 2.87). Indeed, among all the independent variables included in Model 2, Update Bias was correlated only with the difference in memory errors between good and bad news (*Supplementary file 3*). Taking away that variable from the model did not change the results in Model 2 (Beta = 0.680, p=0.001) or Model 1 (Beta = 0.544, p=0.002). The only other variable that was associated with future time in euthymia was whether the patients were taking antidepressants such that taking antidepressants was associated with a shorter time in euthymia (Beta = −0.429; p=0.040). No other clinical predictors were significant (*Supplementary file 2d*).

The update bias has been shown to be related to trait optimism (*Sharot et al., 2011*). We thus examined if trait optimism (measured via the standard LOT-R questionnaire, *Scheier et al., 1994*) was also related to time in euthymia. To that end, we added the LOT-R to Model 2 and found that while the update bias was still related to time in euthymia (Beta = 0.740; p=0.001) LOT-R was not (Beta = 0.208, p=0.148). This suggests that trait optimism is not a predictor of relapse while update bias is.

As the update bias is a predictor of *future* time in euthymia, one may wonder whether it is also related to number of *past* episodes. To test this, we examined the correlation between the two and found no association (r = -.107; p=0.53). Moreover, adding this variable into Model 2 did not change the results (Beta relating update bias to future time in euthymia = 0.717, p=0.005).

The above results indicate that the relationship between update bias and future time in euthymia cannot be accounted for by demographics, clinical indicators of illness course or personality traits. Moreover, as we always control for task variables, the effect cannot be explained by attention or memory of the information received (as these are controlled for by adding memory in the linear regression), or by how familiar or experienced participants were with the stimuli presented (as ratings of these were included in the linear regression), nor by how negative they found the stimuli to be (again, these ratings were included).

## Leave-one-out validation

Next, we tested the predictive validity of the above model using leave one out analysis in which the abovementioned linear regression was ran on all the data save for one participant which was held out from the analysis. We then used the regression betas to predict relapse of the left-out participant. This process was repeated so that each participants' time to relapse was estimated from model parameters generated without using that participant to fit the data. Participants actual time to relapse (data) and their predicted time to relapse (estimation) were then correlated and also compared using a paired sample t-test. This analysis indicates whether the update bias is a good predictor of future time in euthymia.

We found a strong correlation between the predicted future time in euthymia from our model (estimate) and the actual future time spent in euthymia (data) (r = 0.522, p=0.001). The means of the two sets of values were not significantly different from one another (t(34) = −0.178, p=0.859). Running the Leave one out cross-validation analysis using Model 2 but taking out the update bias variable showed that the predicted values and actual values were not correlated (r = 0.129; p=0.459) and model's fit (SSE, sum of squared errors) was better for the model including the Update Bias (SSE = 61.80) than the one that did not (SSE = 97.10). This suggests that the update bias is a crucial variable in a model predicting future time in euthymia.

As Model 2 includes all clinical variables, we wanted to examine whether predictive validity would remain high in a model in which only the task variables were included (that is, the update bias and all task related controls) and was blind to demographics and clinical variables. This leave one out

analysis again showed a significant correlation between predicted future time in euthymia (estimate) and actual future time in euthymia (data) (r = 0.454, p=0.006) with no difference between the means of the two sets of values (t(34) = 0.127; p=0.898).

Overall, our results suggest that a greater update bias predicts a longer time in euthymia. This held true even when controlling for clinical variables.

## Discussion

Our findings show that the extent to which bipolar patients updated their beliefs in response to positive information, relative to negative information, was predictive of when they would relapse. In particular, euthymic patients who updated their beliefs to a greater extent in response to positive information were more likely to remain in euthymia for a longer duration. In other words, biased processing of information in a manner that supports positive outlook was linked to a more favorable course of bipolar illness.

These results are important from a clinical perspective due to the difficulty in predicting the course of bipolar disorder (*Vieta et al., 2018*). Maintenance treatment in bipolar disorder relies on mood stabilizers and psychological programs aimed at increasing treatment compliance, educating the patients and identifying prodromal sub-syndromic symptoms, which provide clues for an upcoming episode (*Oud et al., 2016*; *Scott et al., 2007*). Despite their efficacy, these approaches are highly dependent on the physician's ability to identify and promptly treat symptoms as they arise, to reduce the likelihood of full-blown episodes. Being able to predict a relapse could inform patients and clinicians to step up vigilance to recognize prodromal symptoms and intervene where necessary. The way patients update beliefs could be introduced in the future as a risk prediction tool for bipolar disorder.

Importantly, our results demonstrate that the absence of an optimistic update bias *precedes* the clinical manifestation of relapse. This finding is important for understanding the relationship between valence-dependent learning and mood. While a relationship between mood and valence-dependent learning has been previously demonstrated in depressed patients (*Garrett et al., 2014*; *Korn et al., 2014*), it has been unclear whether a change in mood likely trigger changes in learning or whether changes in learning likely trigger changes in mood. While our data is correlational, the temporal order of events in our study support the latter possibility. That is, we speculate that a reduction in learning in response to positive information relative to negative information over time, lead to a less optimistic perspective, which eventually provides a fruitful ground for clinical affective symptoms to manifest.

It is interesting that a reduction of a positivity bias in belief updating was predictive of both depressive and manic episodes. The results are consistent with multistable models that suggest that environmental or inner perturbation such as stressors could trigger either episode onset (*Cochran et al., 2018*; *Cochran et al., 2017*). The update bias may be an indicator of the existence of such stressors and their associated prodromal symptoms. While mood symptoms of full-blown episode are different for depression and mania, both polarities are preceded by many of the same subsyndromal a-specific symptoms. These symptoms, known as prodromes, represent the main therapeutic target when treating recurrences (*Vieta et al., 2018*) and include sleep disturbances, irritability, and mood instability (*Andrade-González et al., 2020*). It is thus possible that a pessimistic view triggers these common subsyndromal symptoms.

We emphasize that our task carefully controls for an individuals perception of how likely they are to experience the aversive events by including their first estimate of these likelihoods in the model. We also control for how often participants had experienced these events in the past and how familiar they are with the events from others experiencing them. Thus, it is unlikely that the relationship between the update bias and future time in euthymia can be explained by some measure of 'disadvantage setting' of the patients. Indeed, the association between the update bias and time in euthymia was not explained by differences in memory, attention, experience with the negative life events used as stimuli, the perceived negativity of those stimuli, familiarity with the stimuli, or patients' initial predictions of how likely they were to experience the events in the future, as all those were controlled for in our analysis. The association remained strong also when clinical measures, demographics, and other questionnaires were controlled for, such as number of past episodes and trait optimism. Interestingly, no clinical measures were able to predict future time in euthymia,

except for antidepressant medication (*Hooshmand et al., 2019*). Other medication, such as mood stabilizers, did not predict time in euthymia. However, because most of the participants (83.3%) were on mood stabilizers, it is impossible to reliably determine if indeed such medication is associated with time in euthymia and whether taking these medications affects the relationship between task performance and time in euthymia. We also note that the sample size, although in line with the a-priori power analysis, is relatively small for assessing those models with a large number of variables and future studies with a larger sample will be helpful to verify the robustness of the current results.

The fact that general measures of cognitive ability such as memory errors and overall update did not predict time in euthymia suggests that the effect is valence-specific and cannot be explained by a general cognitive impairment, as this would have been better captured by overall update rather than the difference in update between good news and bad news trials. However, it is possible that other valence-dependent measures of learning, such as ones that can be quantified in a reinforcement learning task (*Lefebvre et al., 2017*), may be good predictors of future time of euthymia. This remains to be tested in future studies.

In sum, we found that greater belief updating in response to positive information was predictive of how long euthymic bipolar patients will remain in euthymia. This finding could be informative for developing tools for patient assessment. Future studies should examine whether the same measure is predictive of the reoccurrence of other clinical disorders, most notably depression. Moreover, it would be informative to test at-risk patients to examine if the update bias can be used as a tool for early diagnosis of affective disorders.

# Materials and methods

## Participants

Patients were recruited at the Psychiatric University Clinic of Parma, a public community-based mental health service, from the 31st of January 2013 to the 26th of November 2014. A psychiatrist (PO) evaluated patients using a suitably structured socio-demographic interview to collect anamnestic and data concerning therapy and illness course.

Sample size was based on a previous study (*Korn et al., 2014*), which found a negative correlation between the Update Bias and BDI scores of $r = -0.50$ ($p<0.001$). The required sample size based on this was n = 23 for an alpha of 0.05% and 80% power, beta = 0.80 (*Faul et al., 2007*). As we expected up to 50% dropout over 5 years, we recruited 45 bipolar patients (19 female, 42.2%) aged between 20 and 80 (mean = 45.87; s.d. = 13.09). Fourteen of the patients had performed the belief update task once before for another study approximately 4 months prior, but using different stimuli. No difference emerged between those that performed the task in the past and those that did not.

The study design is observational and prospective. The patients were followed-up with psychiatric visits at the hospital approximately every 2 months (min = 0.25 months, max = 3 months) for 5 years from enrolment (until December 2019) by the same psychiatrist (PO) who is trained in the administration of the SCID-5. The follow-up visits were part of the clinical practice in an outpatients' service specialized in bipolar disorder. The occurrence of new episodes, registered at the time of the visit, was discussed and confirmed by the clinicians according to the DSM-5 criteria. Error could occur if a patient happened to relapse and recovered between the visits (i.e. within the 2-month gap) and the clinician had failed to record these events. This, however, is unlikely as the average duration of a mood episode is 4 months (*Tondo et al., 2017*). Patients for whom there was no contact for a period longer than 6 months were dropped from the study (n = 9) and their data were not included in the survival analysis. This resulted in a final sample of 36 subjects. All participants gave informed consent prior to testing. The local ethics committee (Comitato Etico per Parma) approved the study protocol in accordance with the Helsinki Declaration.

Participants were invited to participate in the study if they: (1) Fulfilled the diagnostic criteria for Bipolar Disorder I or II at the Structured Clinical Interview for DSM IV TR Disorders (*First et al., 2002*) in full remission. (2) Did not satisfy the criteria for Rapid Cycling (i.e. more than four episodes/year) (*American Psychiatric Association, 2013*); (5) Did not have any other major axis I diagnosis (i.e. Schizophrenia spectrum disorder, Obsessive Compulsive Disorder, Panic Disorder, Generalized Anxiety Disorder, Post Traumatic Stress Disorder, Anorexia Nervosa, Binge Eating Disorder). Specific

Phobias and Personality Disorders were not considered because they are outside the study's aim; (6) Did not reveal substance abuse/dependence or addictive disorder in the previous three months; (7) Did not show cognitive impairment, defined as score of less than 25 on the Mini Mental State Examination (*Folstein et al., 1975*).

## Behavioral task

Behavioral task was adapted from our past study (*Sharot et al., 2011*). This task has been used numerous times in the literature (*Chowdhury et al., 2014*; *Garrett et al., 2018*; *Garrett et al., 2014*; *Garrett and Sharot, 2016*; *Garrett and Sharot, 2014*; *Kappes et al., 2018*; *Korn et al., 2014*; *Kuzmanovic et al., 2016*; *Kuzmanovic et al., 2014*; *Ma et al., 2016*; *Moutsiana et al., 2015*; *Moutsiana et al., 2013*; *Sharot et al., 2012b*; *Sharot et al., 2012a*; *Sharot et al., 2011*).

### Stimuli

Forty short descriptions of negative life events (for example: domestic burglary, card fraud) were presented in a random order. The original English stimuli (*Sharot et al., 2011*) were translated into Italian by a native Italian speaker with English as a second language and approved by the author according to a back-translation model. Very rare or very common events were not included; all event probabilities lay between 10% and 70%. To ensure that the range of possible overestimation was equal to the range of possible underestimation, participants were told that the range of probabilities lay between 3% and 77% and were only permitted to enter estimates within this range. Participants were randomly assigned one of three possible lists of stimuli (each list comprised a different set of 40 stimuli).

### Paradigm

All subjects completed a practice session of three trials before beginning the main experiment. On each trial, one of the 40 adverse life events were presented for 4 s. Participants were asked to estimate how likely the event was to happen to them in the future. Participants had up to 6 s to respond. If participants had already experienced an event, they were instructed to estimate the likelihood of that event happening to them again in the future. If the participant failed to respond, that trial was excluded from all subsequent analyses (mean trials with missing response = 4.73, s. d. = 4.03). Following presentation of a fixation cross, participants were presented with the probability of an event occurring in a demographically similar population for 2 s followed by a fixation cross. In a second session, immediately after the first, participants were asked again to provide estimates of their likelihood of encountering the same events so that we could assess how they updated their estimate in response to the information presented. Probabilities of the events occurring were not provided in this second session (*Figure 1*). After the task participants completed the Beck Depression Inventory (BDI-II) (*Beck et al., 1996*) and the Revised Life Orientation Test (LOT-R) to evaluate patients' trait optimism (*Scheier et al., 1994*).

### Memory control

To control for memory effects, participants were asked at the end of the experiment to provide the actual probability previously presented for each event. Memory errors were calculated as the absolute difference between the probability previously presented and the participants' recollection of that statistic:

$$Memory\ Error = |Probability\ Presented - Recollection\ of\ Probability\ Presented|$$

### Other controls

Following previous studies papers adopting the same task in clinically depressed participants (*Garrett et al., 2014*; *Korn et al., 2014*) after completing the task, participants also rated all stimuli on familiarity [for the question ''Regardless if this event has happened to you before, how familiar do you feel it is to you from TV, friends, movies, and so on?'' the responses ranged from 1 (not at all familiar) to 6 (very familiar)], prior experience [for the question ''Has this event happened to you before?'' the responses ranged from 1 (never) to 6 (very often)], vividness [for the question 'How vividly could you imagine this event?' (1, not at all vivid, to 6, very vividly)], emotional arousal [for the question 'When you imagine this event, how emotionally arousing do you find the image in your

mind?' (1, not at all arousing, to 6, very arousing)] and negativity [for the question ''How negative would this event be for you?'' the responses ranged from 1 (not negative at all) to 6 (very negative)].

One subject interrupted the experiment during the last part when memory and other control variables were being registered resulting in missing values for these parameters.

### Statistical analysis

Trials were partitioned into either 'good news' or 'bad news' according to participants' first estimates. A trial was defined as good news when the probability presented was lower than the first estimate of their own probability (*Figure 1a*). Similarly, when the probability presented was higher, the trial was classified as bad news (*Figure 1b*). Trials for which first estimates were equal to the information provided were excluded from subsequent analyses, as these could not be categorized into either condition.

Update was calculated for each trial such that positive updates indicated a change towards the probability presented and negative updates a change away from the probability presented:

$$Update\ (GoodNews) = First\ Estimate\ -\ Second\ Estimate$$

$$Update\ (Bad\ News) = Second\ Estimate\ -\ First\ Estimate$$

We then computed the average update for good and bad news separately for each participant, as done previously (*Chowdhury et al., 2014*; *Garrett et al., 2018*; *Garrett et al., 2014*; *Garrett and Sharot, 2014*; *Kappes et al., 2018*; *Korn et al., 2014*; *Kuzmanovic et al., 2016*; *Kuzmanovic et al., 2014*; *Moutsiana et al., 2015*; *Moutsiana et al., 2013*; *Sharot et al., 2012a*; *Sharot et al., 2012b*; *Sharot et al., 2011*). A participant's update bias was then computed as the signed difference between these two scores:

$$Update\ Bias = Average\ Update\ (Good\ News) - Average\ Update\ (Bad\ News)$$

A score of 0 indicates no bias in updating in either direction; positive scores indicate an optimistic bias in updating such that participants change their beliefs to a greater degree for good news relative to bad news. Negative scores indicate a pessimistic bias in updating such that participants change their beliefs to a greater degree for bad news relative to good news.

This bias can be seen both when trials are classified according to the participants' estimate of base rate or self-risk (*Garrett and Sharot, 2014*; *Kuzmanovic et al., 2014*). Similarly, this bias is not affected by the estimate being either the likelihood of the event happening or not happening to them (*Garrett et al., 2014*; *Garrett and Sharot, 2014*; *Sharot et al., 2011*). Hence, here we use the classic approach in which we elicited the estimation of an event happening to them.

We also computed the differences between good and bad trials on all the other task related variables, that were included as covariates in all the following models.

$$\Delta(X) = Average\ X_{good\,news} - Average\ X_{bad\,news}$$

This deltas resulted in a set of new variables named $\Delta$ estimation error, $\Delta$ memory errors, $\Delta$ number of trials, $\Delta$ reaction times at the first and second estimate, $\Delta$ familiarity, $\Delta$ prior experience, $\Delta$ vividness, $\Delta$ emotional arousal and $\Delta$ negativity.

### Future time in euthymia

For each subject, we calculated the time in months they spent in euthymia following task completion until the onset of the next episode up to 5 years. For those who did not relapse during the follow-up, time in euthymia was set to 60 months (n = 6).

Bipolar disorder is a recurrent disorder with a mean distance between episodes of approximately 1.5 years with half of the patients relapsing in the following year (*Radua et al., 2017*). As future time in euthymia was not normally distributed (Skewness = 1.362; Kolmogorov-Smirnov normality test = 0.301, p<0.001), we log-transformed this variable resulting in a normal distribution of the values (Skewness = −0.338; Kolmogorov-Smirnov normality test = 0.118; p>0.05).

## Update bias and future time in euthymia

We conducted a linear regression with future time in euthymia as the dependent variable and update bias as our predictor of interest. To control for possible confounds, we also added as independent variables the mean first estimate and the difference between good news trials and bad news trials on the following measures: estimation errors, memory error, number of trials, reaction times at the first and second estimate and the subjective ratings on familiarity, prior experience, vividness, emotional arousal and negativity. We also ran this regression with bootstrapping in SPSS (10,000 runs with replacement).

### Model 1

Log (Future time in euthymia)=b0 + b1* update bias + b2* mean first estimate + b3* $\Delta$ estimation error + b4* $\Delta$ memory errors + b5* $\Delta$ number of trials + b6* RT first estimate + b7* RT second estimate + b8* $\Delta$ familiarity +b9* prior experience +b10* $\Delta$ vividness + b11*$\Delta$ emotional arousal + b12* $\Delta$ negativity + error.

$\Delta$ (delta) refers to the difference in each parameter between good and bad news trials.

The same linear regression was run two more times substituting the update bias with the update in response to good news and the update in response to bad news. To visualize the effect of each of these (see *Figures 2*, *3* and *4*), we generated scatterplots of the residuals of the dependent variable (logarithmic transformation of the future time in euthymia in months to account for skewness) and of the independent variable of interest (update bias, update from good news, update from bad news respectively), this accounts for the other control variables.

Because it is possible that the update bias predicts future time in euthymia differently for manic and depressive episodes, we added to the linear regression above the polarity of the next episode and its interaction with the update bias as predictors. Six subjects did not relapse during the follow-up period, hence this multiple linear regression was run on 30 subjects only. For this analysis, next episode polarity was coded as −1 for depressive episodes and one for manic episodes.

To test the strength of this model, we ran model averaging (*Freckleton, 2011*). Model averaging tests whether the effect of the update bias was contingent on entering a specific set of variables in the model (*Fletcher, 2018*). This approach involves running every single combination of models given the independent variables. For example, a multiple linear regression with three predictors (a, b, c) will results in seven different models: three with one predictor each (a, b, and c), three with the possible combination of two predictors (a + b, a + c, and b + c) and one full model with the three predictors (a + b + c). The betas of each predictor are then averaged across all models, weighting them on the model's BIC (Bayesian Information Criterion). Moreover, the nested models can be ordered according their BIC score, where a lower BIC corresponds to a better fit. This allows to evaluate which independent variables are most relevant for the full model.

Because difference between good news trials and bad news trials in task-related variables and subjective ratings might introduce noise when evaluating the association between the update bias and the time in euthymia we controlled for those in a multiple linear regression. To identify these variables, we tested if any of the variables included in Model 1 showed significant differences between good and bad news trials using a paired sample t-test (e.g. compared memory error scores for good news versus memory error scores for bad news, vividness ratings for good news versus bad news, and so on for all variables in Model 1). We then ran a linear regression with future time in euthymia as the dependent measure and update bias (our main measure of interest) as the independent measure controlling only for the measures that returned a significant results from the above set of tests.

To control for all demographic and clinical variables, we then ran a second linear regression with the following controls: age, education, gender, bipolar type (I or II), in-range plasmatic dosage of mood stabilizer (lithium, valproate, lamotrigine and carbamazepine), antipsychotics or antidepressant prescription, history of psychotic symptoms, duration of illness, and the BDI-II scores.

### Model 2

Log (Future time in euthymia)=b0 + b1* update bias + b2* mean first estimate + b3* $\Delta$ estimation error + b4* $\Delta$ memory errors + b5* $\Delta$ number of trials + b6* RT first estimate + b7* RT second estimate + b8* $\Delta$ familiarity + b9* prior experience + b10* $\Delta$ vividness + b11* $\Delta$ emotional arousal +

b12* Δ negativity + b13* age + b14* years of education + b15* gender + b16* bipolar type + b17* depressive symptoms at BDI-II + b18* mood stabilizers + b19* antidepressants + b20* antipsychotics + b21* lithium + b22* history of psychotic symptoms + b23* duration of illness + error.

Δ (delta) refers to the difference in each parameter between good and bad news trials; gender coded as 0 = male, 1 = female; mood stabilizers, antidepressants, antipsychotics, lithium and history of psychotic symptoms coded as 0 = no, 1 = yes.

To test for multicollinearity, using the SPSS collinearity diagnostics we calculated the Variance Inflation Factor (VIF) for our variable of interest. The VIF quantifies the amount of multicollinearity in a set of regression variables. It is equal to the ratio of the overall model variance to the variance of a model that includes only that single independent variable. A high VIF indicates that the independent variable is highly collinear with the other variables in the model. A low VIF (less than 4) indicated it is not.

### Leave-one-out validation

Next, we tested the robustness of the Model 2 using leave one out cross validation. Model 2 was rerun on all data except for one subject. The regression betas are then used to predict the relapse of that left out participant. This process was repeated for each participant so that the leave-one-out reiteration resulted in two values for time in euthymia for each subject: the actual time in euthymia (data) and the predicted value (estimate). We then ran a Pearson correlation between the two values as well as compared them using a t-test.

Finally, we performed the same leave one out analysis as above but this time *without* including the Update Bias. A non-significant correlation between the actual and predicted values of this new model without the update bias would suggest a key-role of our variable of interest in predicting time in euthymia.

## Additional information

### Funding

| Funder | Grant reference number | Author |
| --- | --- | --- |
| Ministry of Education, University and Research | D.R.LXXXIII-887-21/05/2014 | Paolo Ossola |
| Wellcome Trust | Senior Research Fellowship 214268/Z/18/Z | Tali Sharot |
| Sir Henry Wellcome Postdoctoral Fellowship | 209108/Z/17/Z | Neil Garrett |

The funders had no role in study design, data collection and interpretation, or the decision to submit the work for publication.

### Author contributions

Paolo Ossola, Conceptualization, Resources, Data curation, Formal analysis, Investigation, Methodology, Writing - original draft, Project administration; Neil Garrett, Conceptualization, Data curation, Software, Formal analysis, Methodology, Writing - review and editing; Tali Sharot, Conceptualization, Supervision, Validation, Methodology, Writing - original draft, Writing - review and editing; Carlo Marchesi, Conceptualization, Resources, Data curation, Supervision, Funding acquisition, Methodology, Project administration, Writing - review and editing

### Author ORCIDs

Paolo Ossola https://orcid.org/0000-0002-0644-3158
Tali Sharot http://orcid.org/0000-0002-8384-6292

### Ethics

Human subjects: All participants gave informed consent prior to testing. The local ethics committee (Comitato Etico per Parma) approved the study protocol (protocol number 45411 approved the 7th of December 2012) and the study has been conducted in accordance with the Helsinki Declaration.

## Decision letter and Author response

Decision letter https://doi.org/10.7554/eLife.58891.sa1
Author response https://doi.org/10.7554/eLife.58891.sa2

## Additional files

### Supplementary files

• Supplementary file 1. Sample descriptive (a) and task-related variables (b).

• Supplementary file 2. Regression coefficients for Model 1 when the predictor of interest are the update bias (a), update from good news (b) and update from bad news (c), and for Model 2 (d).

• Supplementary file 3. Correlations among the independent variables included in the Models.

• Transparent reporting form

### Data availability

All data generated and analysed during this study are included in the manuscript and supporting files. Specifically the complete dataset is available on Mendeley Data.

The following dataset was generated:

| Author(s) | Year | Dataset title | Dataset URL | Database and Identifier |
|---|---|---|---|---|
| Ossola P | 2020 | Belief Updating in Euthymic Bipolar Disorder | https://doi.org/10.17632/crp8k3dgj5.2 | Mendeley Data, 10.17632/crp8k3dgj5.2 |

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
