## [Decision Letter]

**Acceptance summary:**

This research highlights a predictor of mood states in individuals with bipolar disorder, above and beyond clinical predictors. Namely, a bias in belief updating when an initial prediction overstates prevalence (i.e. "good news" situation) relative to when an initial prediction understates an outcome (i.e. "bad news" situation) is predictive of future time spent euthymic in individuals with bipolar disorder. These findings could lead to novel therapeutic interventions for those with bipolar disorder.

**Decision letter after peer review:**

Thank you for submitting your article "Belief Updating in Bipolar Disorder Predicts Time of Recurrence" for consideration by *eLife*. Your article has been reviewed by three peer reviewers, and the evaluation has been overseen by Shelly Flagel as a Reviewing Editor and Michael Frank as the Senior Editor. The following individuals involved in review of your submission have agreed to reveal their identity: Amy Louise Cochran (Reviewer #1); Catherine Harmer (Reviewer #3).

The reviewers have discussed the reviews with one another and the Reviewing Editor has drafted this decision to help you prepare a revised submission.

The editors have judged that your manuscript is of interest, but as described below noted that additional analyses and significant revisions are required before it is published. Thus, we would like to draw your attention to changes in our revision policy that we have made in response to COVID-19 (https://elifesciences.org/articles/57162). First, because many researchers have temporarily lost access to the labs, we will give authors as much time as they need to submit revised manuscripts. We are also offering, if you choose, to post the manuscript to bioRxiv (if it is not already there) along with this decision letter and a formal designation that the manuscript is "in revision at *eLife*". Please let us know if you would like to pursue this option. (If your work is more suitable for medRxiv, you will need to post the preprint yourself, as the mechanisms for us to do so are still in development.)

Summary:

Ossala et al. seek to predict future time spent euthymic in individuals with bipolar disorder (BP) based on how they update their beliefs. To predict time spent euthymic, they emphasize one variable, in particular, the bias in average updates when their initial prediction overstates actual prevalence (which they call a "good news" situation) vs. when their initial prediction understates actual outcomes (which they call a "bad news" situation). The principal finding is that the bias in average updates is predictive of future time spent euthymic after controlling for a number of possible confounds including subjective biases, task-related variables, and clinical variables. The study design is thoughtful and interesting, and the principal finding appears to be exceedingly strong. If biases in updating is a robust predictor of future outcomes, above and beyond clinical predictors, then this variable should have utility in helping set expectations for individuals with BP as well as helping get at underlying cognitive deficits in BP.

Essential revisions:

Considering the small sample size (n = 36 outcomes) and the number of covariates (e.g., n = 24 in Model 3), a major concern is the possibility of an overinflation of p values for various reasons given below. Based on this concern, the authors need to address the following:

1) Did the authors perform a power analysis before study start? The small sample size should be raised as a limitation in the Discussion.

2) There may be multicollinearity among independent variables in the regression models that might lead to an overinflation of p values. Broadly, please address the issue of multicollinearity. Specifically, can you report univariate analyses relating update bias to time spent euthymic? And can you report (in the supplement) the correlation between independent variables in the regression model?

3) While cross-validation was used to show the prediction models would generalize to a larger population, it was not used to determine if the principal finding (i.e. relationship between update bias and time spent euthymic) would generalize. Can you use cross-validation to evaluate the generalizability of this finding, e.g., by comparing cross-validated prediction errors to a model that does not include update bias?

4) Based on the Materials and methods section, it appears that you are running a standard linear regression, but you mention that you construct general linear models (GLM). If the former, please simply state that. If the latter, can you provide more details in what exact analysis was performed? For example, using repeated measures could lead to lower p values, but would need to be described (though inappropriate for the given outcome which is only measured once per individual).

5) Please discuss the small sample size as a limitation in the Discussion. Assuming a standard linear regression was performed, there should be 5-10 outcomes per independent variable, but the presented analysis only had 2-3.

6) Given that the task has been analyzed elsewhere, please compare/contrast your analyses to analyses from prior papers? Among task-related variables (e.g., difference in number of good vs. bad trials) that were controlled for in the regression models, how many were used in prior papers?

7) Related to the last comment, in prior work using the task and in models of associative learning more broadly, it is common to analyze bias in learning rates (roughly the ratio between the update and the error) rather than the update bias. Can the authors comment on why they focused on bias in overall updates rather than bias in learning rates?

8) One might have expected that optimistic updates predict the transition only to manic and not depressives states. The authors should discuss this and other alternative hypotheses and whether their reported findings fit into any existing and coherent model of BP disorder.

9) The authors show that the belief update task can predict the time of the dysthymic transition. But they only have tested the belief update task. What if any task predicts the time of the dysthymic? On the other side, optimistic biases can be assessed in several ways. On one side of the spectrum there are standard personality measures, such as the LOT-R, on the other side of the spectrum there are reinforcement learning tasks, where the optimistic bias is frequently reported. Does any measure of optimism predict the time of the dysthymic transition? Or is it just the belief update task? In the Supplementary material it is mentioned that trait optimism (LOT-R) was assessed. Does this scale also predict the time spent in euthymia? Do the results hold if it is included in the GLMs?

10) Another important check that is lacking is whether the results hold if the number of previous dysthymic episodes is entered as a control. In such pathological states, it seems that crises become more and more frequent with time. A possible interpretation of the results (that cannot be ruled out presently) is that a higher number of previous dysthymic episodes makes a patient concomitantly less optimistic and more likely to have a shorter euthymic phase. A related question is whether the belief task is a better predictor compared just the number of previous dysthymic episodes (note that even if correlated this factor is not completely taken into account by including the patient age in the regression).

11) Medication effects were controlled for statistically by including 0 or 1 for different classes of medication. However, this is unlikely to be a perfect control as few participants are medication free (>80% on a mood stabiliser). This should be discussed in more detail.

12) The inclusion of the different clinical and demographic factors that might influence the association is appreciated. One might wonder if this could just be a general cognitive effect e.g. the more “impaired” cognition (or difference to normal) associated with a worst illness profile. The authors go some way towards assessing this by looking for example at memory for the new information presented in the task. However, this justifies a bit more attention i.e. did the authors include relapse history in the model? Was memory (as oppose to belief updating) a predictor by itself?

13) What is the likely mechanism underpinning the failure to update beliefs on the basis of positive information? Is it that despite knowing the population average, the participant believes they have more specific information to refute that a positive update applies to them e.g. burglaries happen to 20% of people. but for some its 30% because they live in an area where burglaries are more common. So, could it be that indirectly the task measures the disadvantaged setting of individuals more prone to relapse? Or is it in the mechanism by which this learning takes place?

---

## [Author Response]

Essential revisions:Considering the small sample size (n = 36 outcomes) and the number of covariates (e.g., n = 24 in Model 3), a major concern is the possibility of an overinflation of p values for various reasons given below. Based on this concern, the authors need to address the following:1) Did the authors perform a power analysis before study start? The small sample size should be raised as a limitation in the Discussion.

Sample size was based on a previous study (Korn et al., 2014), which found a negative correlation between the Update Bias and BDI scores of r = -0.50 (p < 0.001). The required sample size based on this was n=23 for an alpha of 0.05 and 80% power, beta=0.80 (Faul et al., 2007). We started with an N of 45 and following drop out over the five years ended with an N of 36. We now include this information in the Materials and methods section. We have also added a Discussion of the small sample size.

2) There may be multicollinearity among independent variables in the regression models that might lead to an overinflation of p values. Broadly, please address the issue of multicollinearity. Specifically, can you report univariate analyses relating update bias to time spent euthymic? And can you report (in the supplement) the correlation between independent variables in the regression model?

To test for multicollinearity, we calculated the Variance Inflation Factor (VIF) for each independent variable. The VIF quantifies the amount of multicollinearity in a set of regression variables. It is equal to the ratio of the overall model variance to the variance of a model that includes only that single independent variable. A high VIF indicates that the independent variable is highly collinear with the other variables in the model. A low VIF (less than 4) indicates that it is not. The VIF of our measures of interest – the update bias – was 1.93 for model 1 and 2.87 for model 2, which is low and not considered problematic. We now report this in the revised manuscript.

Indeed, the only independent variable the update bias was significantly correlated with was difference in memory errors. Taking away that variable from the model did not change the results (Update bias Beta = 0.544, p = 0.002 for model 1; Beta = 0.680, p = 0.001 for model 2). We now report the correlation between all independent variables in the regression models as requested (see Supplementary Material).

We also tested whether the effect of the update bias was contingent on entering a specific set of variables in the model by running “model averaging” (Freckleton, 2011). This approach involves first running every single combination of models given the independent variables. For example, running a model only with 2 of the variables, only 3, only 4 and so on. Each time with a different combination of independent variables. With N = 13 (Model 1) this involved running 4095 nested models. Then the betas of each variable are averaged across all models, weighting them on the model’s BIC (Bayesian Information Criterion). This revealed a significant effect of update bias in explaining “time in euthymia” (weighted estimates = 0.47, 95% CI = 0.34-0.60). The eight best fitting models (out of 4095), according to the lowest BIC score, all included the update bias. Running the same procedure for Model 2 will involve running 8,388,607 models and we lack the computational power needed for that exercise. However, given the significant effects of Model 1, the goal of Model 2 is simply to ensure that the results of Model 1, which includes less variables, are not explained away by additional clinical and demographic variables.

While multicollinearity is not driving the effects, our additional analysis reveals that there are differences between the two types of trials (good news trials and bad news trials) that introduce noise and can obscure the effects of update bias if not controlled for. In particular, testing if any of the task variables showed significant differences between desirable and undesirable trials revealed significant effects in all subjective ratings: past experience, familiarity, vividness, arousal, vividness and negativity (p-values < 0.02) but not in any of the other variables (that is no difference for memory errors, estimation errors, number of trials or RTs). Simply correlating update bias with time in euthymia without accounting for these significant differences reveals a trend (rho = 0.297 p = 0.078), and once the differences are accounted for by running a new model controlling only for these subjective measures the relationship between Update Bias and “time in euthymia” is clearer (Beta = 0.351; p = 0.047). The VIF of the update bias in this model is low (VIF = 1.098) as were the VIFs of all other variables (all VIFs < 4). We now report all this in the revised manuscript.

3) While cross-validation was used to show the prediction models would generalize to a larger population, it was not used to determine if the principal finding (i.e. relationship between update bias and time spent euthymic) would generalize. Can you use cross-validation to evaluate the generalizability of this finding, e.g., by comparing cross-validated prediction errors to a model that does not include update bias?

Thank you for suggesting this confirmatory analysis. Running the LOO-cv analysis using the model that does not include update bias showed that the predicted values and actual values were not correlated (r = 0.129; p = 0.459). Compare this to the same model which does include update bias, for which we find both a strong correlation between predicted values and actual values (r = 0.445, p = 0.007) and no significant difference between these two values (t(34) = 0.123; p = 0.901, paired sample t-test). The model’s fit (SSE, sum of squared errors) was better for the model including the Update Bias (SSE = 61.80) than the one that did not (SSE = 97.10). We have now added this analysis to the revised manuscript.

4) Based on the Materials and methods section, it appears that you are running a standard linear regression, but you mention that you construct general linear models (GLM). If the former, please simply state that. If the latter, can you provide more details in what exact analysis was performed? For example, using repeated measures could lead to lower p values, but would need to be described (though inappropriate for the given outcome which is only measured once per individual).

Thank you for noting this. We ran linear regressions not general linear models. This typo has now been corrected.

5) Please discuss the small sample size as a limitation in the Discussion. Assuming a standard linear regression was performed, there should be 5-10 outcomes per independent variable, but the presented analysis only had 2-3.

Following the reviewer’s suggestion, we have now added a discussion of the limitation of the small sample size. In addition, we now perform two additional analyses that help alleviate the concern of number of outcomes per independent variable. First, we rerun the regression analysis using bootstrapping which enables us to construct confidence intervals with a small sample. Performing bootstrapping on 10000 samples again showed that the relationship between the Update Bias was significantly related to time in Euthymia (Beta = 0.589, CI = 0.122-1.129). Second, we run model averaging (Fletcher, 2018). In particular, running 4095 nested models, most of which include a smaller number of independent variables. This again revealed a significant effect of update bias in predicting “time in euthymia” (Estimates weighted by log model evidence = 0.465, 95% CI = 0.34-0.60). Third, we now report the results of a new model with only 6 variables, which is 6 outcomes per independent variable. We now add these additional analyses to the revised manuscript.

6) Given that the task has been analyzed elsewhere, please compare/contrast your analyses to analyses from prior papers? Among task-related variables (e.g., difference in number of good vs. bad trials) that were controlled for in the regression models, how many were used in prior papers?

Our approach was to include all the control variables that were included in the previous papers adopting the same task in clinically depressed participants (Korn et al., 2014; Garrett et al., 2014). We now mention this in the Materials and methods. To our knowledge, seven past papers (Chowdury et al., 2013; Korn et al., 2014, 2016; Garrett et al., 2014, 2018; Kuzmanovic et al., 2017; Ma et al., 2016) include difference in number of good vs. bad trials as controls, and nine papers include RT (Sharot et al., 2011; Sharot et al., 2012b, Chowdury et al., 2013; Korn et al., 2014; Garrett et al., 2014; Moutsiana et al., 2013; Kuzmanovic et al., 2017, 2018; Ma et al., 2016). All papers from the Affective Brain Lab (N = 10) include memory and subjective ratings (i.e. familiarity, prior experience, vividness, emotional arousal and negativity) as controls, as do some papers not from the lab (e.g., Ma et al., 2016; Oganiana et al., 2018; Schonfelder et al., 2017). Thus, the controls included in the paper are in line with past studies.

7) Related to the last comment, in prior work using the task and in models of associative learning more broadly, it is common to analyze bias in learning rates (roughly the ratio between the update and the error) rather than the update bias. Can the authors comment on why they focused on bias in overall updates rather than bias in learning rates?

We focused on the update itself rather than the ratio between update and error for the following reasons. First, in the 20 papers published thus far with the update task 100% used update as a measure of interest (Sharot et al., 2011, 2012a, 2012b; Chowdury et al., 2013; Korn et al., 2014; Moutsiana et al., 2013, 2015; Garrett et al., 2014, 2017, 2018; Kappes et al., 2018; Oganian et al., 2018; Kuzmanovic et al., 2014, 2016, 2017, 2018, 2019; Ma et al., 2016; Schonfelder et al., 2017) and only 40% use the ratio between update and error as a measure of interest (Sharot et al., 2011; Kuzmanovic et al., 2017, 2018; Kappes et al., 2018; Ma et al.,l 2016). Moreover, in the three papers that use this task to examine affective disorders all three used update as a measure of interest (Garrett et al., 2014; Korn et al., 2014; Schonfelder et al., 2017) and only one used the ratio between update and error as a measure of interest. Thus, to be consistent with past research we focused on the update measure. Second, the reason that past studies, in particular those looking at patients (Garrett et al., 2014; Korn et al., 2014, 2016; Kuzmanovic et al., 2019), focus on update rather than the ratio between update and error is that such populations usually show a general impairment in the latter but not the former. This makes the ratio a noisy measure to use in such populations. Indeed, the average ratio between update and error for good news in our clinical sample was 0.63, below that seen in healthy populations (0.72 in the original study by Sharot et al., 2011; and ranging between 0.67 in Kappes et al., 2018 to 0.77 in Ma et al., 2016).

8) One might have expected that optimistic updates predict the transition only to manic and not depressives states. The authors should discuss this and other alternative hypotheses and whether their reported findings fit into any existing and coherent model of BP disorder.

Thank you for raising this question. Indeed, one hypothesis would be that greater update bias will be associated specifically with transition to a manic episode. This type of prediction will be compatible with the notion held by some that mood is a one-dimensional construct with depression and mania as opposites of the same continuum (Chang and Chou, 2018). An alternative hypothesis (which our data supports) is that greater update bias will be associated with episodes of either polarity. Such a hypothesis would be consistent with multi-stable models that suggest that environmental or inner perturbation such as stressors could trigger either episode onset (Cochran et al., 2017, 2018). The update bias may be an indicator of the existence of such stressors and their associated prodromal symptoms. These models consider mood a two-dimensional construct and suggest that mania and depression can vary independently and do not inhibit each other. We now detail these hypotheses in the revised manuscript.

9) The authors show that the belief update task can predict the time of the dysthymic transition. But they only have tested the belief update task. What if any task predicts the time of the dysthymic? On the other side, optimistic biases can be assessed in several ways. On one side of the spectrum there are standard personality measures, such as the LOT-R, on the other side of the spectrum there are reinforcement learning tasks, where the optimistic bias is frequently reported. Does any measure of optimism predict the time of the dysthymic transition? Or is it just the belief update task? In the Supplementary material it is mentioned that trait optimism (LOT-R) was assessed. Does this scale also predict the time spent in euthymia? Do the results hold if it is included in the GLMs?

Following the reviewer’s comment, we tested whether the LOTR was associated with time in euthymia. It was not (rho = 0.164, p = 0.340). Moreover, including the LOT-R in our model showed that the results hold. That is, update bias was still related to time in euthymia (beta = 0.740; p = 0.001) while LOTR was not (beta = 0.208, p = 0.148). In addition, we note that other measures such as memory bias (that is memory errors for positive trials minus for negative trials) also did not predict time in euthymia (beta = 0.13; p = 0.959). It may certainly be that other measures not tested here, such as a bias in learning in a reinforcement learning task (e.g., Lefebvre et al., 2017) may be related to time in euthymia. That will require future testing. However, our results show that not *all* tasks/measures predict time in euthymia. We now include this information in the revised manuscript.

10) Another important check that is lacking is whether the results hold if the number of previous dysthymic episodes is entered as a control. In such pathological states, it seems that crises become more and more frequent with time. A possible interpretation of the results (that cannot be ruled out presently) is that a higher number of previous dysthymic episodes makes a patient concomitantly less optimistic and more likely to have a shorter euthymic phase. A related question is whether the belief task is a better predictor compared just the number of previous dysthymic episodes (note that even if correlated this factor is not completely taken into account by including the patient age in the regression).

We thank the reviewer for prompting us to examine the effect of number of previous episodes. We extracted the number of previous episodes from the clinical records and found that it was not associated with time in euthymia (r = -0.013; p = 0.93) nor with the Update Bias (r = -0.107; p = 0.53). Adding this variable into the model did not change the results: update bias was still associated with time in euthymia (Beta = 0.066, p = 0.013). Thus, we conclude that update bias is a better predictor than number of previous dysthymic episodes in predicting future time in euthymia. We now add this information to the manuscript.

11) Medication effects were controlled for statistically by including 0 or 1 for different classes of medication. However, this is unlikely to be a perfect control as few participants are medication free (>80% on a mood stabiliser). This should be discussed in more detail.

It is true that given that most participants were on mood stabilisers it is impossible to reliably determine the effect of medication on the relationship between task performance and time in euthymia. This question could only be addressed if there are sufficient patients not on such medication. We now highlight this caveat in the Discussion.

12) The inclusion of the different clinical and demographic factors that might influence the association is appreciated. One might wonder if this could just be a general cognitive effect e.g. the more “impaired” cognition (or difference to normal) associated with a worst illness profile. The authors go some way towards assessing this by looking for example at memory for the new information presented in the task. However, this justifies a bit more attention i.e. did the authors include relapse history in the model? Was memory (as oppose to belief updating) a predictor by itself?

Following the reviewers’ suggestion, we looked at the association between the overall magnitude of memory errors and future time in euthymia and found no association (r = -0.008; p = 0.964). Neither was relapse history, calculated as the number of previous episodes, associated with future time in euthymia (r = -0.013; p = 0.939). Finally, we note that the association between update and time in euthymia is valence-dependent. Greater update in response to positive information is associated with longer time in euthymia but more update in response to negative information is associated with less time in euthymia. We thus conclude that the association between the update bias and time in euthymia is unlikely to be explained by a general cognitive impairment as this would have been better captured by overall update and overall memory performance. We now add this to the revised manuscript.

13) What is the likely mechanism underpinning the failure to update beliefs on the basis of positive information? Is it that despite knowing the population average, the participant believes they have more specific information to refute that a positive update applies to them e.g. burglaries happen to 20% of people. but for some its 30% because they live in an area where burglaries are more common. So, could it be that indirectly the task measures the disadvantaged setting of individuals more prone to relapse? Or is it in the mechanism by which this learning takes place?

We thank the reviewer for raising this question. Our task carefully controls for people’ perception of how likely they are to experience these aversive events by including their first estimate of these likelihoods in the model. We also have two additional measure of the “disadvantaged setting” of the participants; we ask them how often they had experienced these events in the past and how familiar they are with the events from others experiencing them. Including the average rating on either of these measures in our model did not change the main result: update bias was still associated with future time in euthymia (Beta = 0.806; p = 0.003) while these measures were not (past experience Beta = -0.022 p = 0.929; familiarity Beta = 0.177 p = 0.416). It is possible that the failure to update in response to positive information is due to the participant perceiving the positive information as irrelevant to them.